# Spatiotemporal Characteristics of Meteorological and Agricultural Droughts in China: Change Patterns and Causes

Lusheng Li , Lili Zhao and Yanbin Li *

School of Water Conservancy, North China University of Water Resources and Electric Power, Zhengzhou 450046, China
* Correspondence: liyanbin@ncwu.edu.cn; Tel.: +86-0371-65790417

**Abstract:** Drought is complex and is also one of the main disasters affecting China. Exploring the response of agricultural drought and meteorological drought to climatic factors helps us to understand the causes of drought. In this paper, we evaluated the temporal and spatial characteristics of soil drought and meteorological drought (SMD) and explored their responses to climatic factors and latent heat fluxes (LHF), and then explained their variation from the perspective of atmospheric circulation. The following results were obtained. (1) Meteorological drought has gradually increased in the Liaohe River Basin, on the North China Plain, and on the Loess Plateau while average soil moisture has been maintained at only approximately 25%. The impacts of drought are very serious in these regions. (2) LHF response to short time-scale (3-month scale) drought performance is high in the dry season, and the regions with high correlation coefficients are spatially distributed and concentrated in the monsoon climate zone. The regions with high correlation coefficients between drought and LHFs on long time scales (12-month scale) are concentrated in the coastal basin of southeast China. (3) Short- and long-term SMDs showed highly responsive and significant relationships with PDO, showing variations in the southeast coastal basin, the Pearl River basin, the northwest inland basin and the eastern part of the Heilongjiang basin, with a maximum correlation coefficient of 0.21 ($p < 0.01$). The short-term SMD in the northwestern inland region was significantly negatively correlated with AMO (correlation coefficient of $-0.19$, $p < 0.01$). the Nino3.4 index is significantly positively correlated with the SMD in the southeast coastal region of China, with a maximum correlation coefficient of 0.23 ($p < 0.01$). The decrease in convective precipitation led to a stronger association between soil and meteorological drought and climatic factors. This study helps to reveal the changing patterns of SMDs and can also be used globally to identify the local development patterns of drought under climate change.

**Keywords:** soil drought; meteorological drought; climatic factors; China

## 1. Introduction

Drought is the most extensive and harmful extreme climate event and is a very destructive kind of natural disaster. Drought disasters have seriously influenced agriculture in China and have threatened the safe production of grain [1,2]. Meteorological drought usually involves insufficient precipitation with increased atmospheric evaporative demand (e.g., due to high temperatures, high solar radiation, or high winds). Soil drought is considered to have set in when the soil moisture availability to plants has dropped to such a level that it adversely affects the crop yield and hence agricultural profitability [3]. Several studies have shown the climate–catchment–soil control on hydrological droughts and identify key drought drivers (KDD) for drought propagation [4]. Meteorological and agricultural droughts usually combine to affect vegetation [5]. Soil drought and meteorological drought are greatly exacerbated by land–atmosphere feedbacks.

Agricultural disasters are induced under the effect of meteorological drought and soil drought. Such disasters are manifested by a decrease in photosynthesis and transpiration of

plants and a decrease in plant productivity [6–8]. Some research has shown that the negative connection between temperature and droughts encompasses most of China, indicating that high temperatures and droughts are generally concurrent in these regions [9]. Drought was most widespread and sustained in the late 1990s and early 2000s, primarily in the southern Yellow River Basin and the northern Yangtze River Basin. [10]. Precipitation and potential evapotranspiration are projected to increase throughout China [11].

Soil drought and meteorological drought (SMD) often trigger forest fires, insect infestations, and socioeconomic impacts [12,13]. The normal growth and development of vegetation can be affected when the soil moisture is below 25% [14].

This SMD leads to vegetation and ecosystem damage, and, through feedback mechanisms, drought leads to increased temperatures and decreased precipitation, thus further enhancing the intensity of the joint drought effects [15–18]. Droughts are also very closely related to food disasters. A drought vulnerability and food security framework has been built based on MaxEnt (maximum entropy) and ANN (analytical neural network) [19]. Vector machines (SVM), random forests (RF), and support vector regression (SVR) are used for planning in drought-prone areas [20].

There are many indices used for soil and meteorological drought monitoring [21–27]. In this study, we choose the standardized precipitation evapotranspiration index (SPEI) and standardized soil moisture index (SSMI) to analyze the co-drought phenomenon. The SPEI can detect drought in a climate-change environment and can respond to climate change better than the previously utilized drought indices [24]. The SSMI is a soil drought index constructed based on soil moisture time series; this index can be used to detect agricultural drought [24–27].

Climate factors are indicative of drought, so exploring the relationship between climate factors and soil and meteorological drought can provide meaningful guidance for drought management. Modern climatologists have focused on the pronounced effects of internal variabilities (e.g., the El Niño–Southern Oscillation (ENSO) the Atlantic Multidecadal Oscillation (AMO), and the Pacific Decadal Oscillation (PDO), and other factors) on the climate system [28]. The ENSO, PDO and AMO [29,30] affect the dry and wet conditions in the inland areas of China [31–34].

Considering these research gaps, the objectives of this study are: (1) Exploring temporal and spatial correlation coefficients between climatic factors and SMDs in dry and non-dry months; (2) to explore the response of drought at different time scales to climate factors. We explore the temporal response of SMDs to climate factors based on the above objectives. The key objective of this article is to identify the impact of climate change on SMDs. These insights provide a reference for drought preparedness and shaping policy recommendations for agricultural and industrial sectors.

## 2. Materials and Methods

### 2.1. Materials

In this study, we used Climatic Research Unit (CRU) 4.0 meteorological data (temperature, precipitation, humidity, and wind speed) obtained at a spatial resolution of $0.5° \times 0.5°$ with a time series length of 1901–2018 (https://lr1.uea.ac.uk/cru/data (accessed on 5 September 2022)) [35]. The soil moisture data were collected from the Global Land Evaporation Amsterdam Model (GLEAM) (https://www.gleam.eu/ (accessed on 5 September 2022)) with a time span of 1980–2018 and a spatial resolution of $0.25° \times 0.25°$ [25,36]. The Pacific Decadal Oscillation (PDO), Atlantic Multidecadal Oscillation (AMO), and Nino3.4 (East Central Tropical Pacific SST) climate factors were obtained from the National Oceanic and Atmospheric Administration (NOAA) Physical Sciences Laboratory of the National Weather Service (https://psl.noaa.gov/data/climateindices./list/ (accessed on 5 September 2022)). It has been shown that the Nino3.4 is used to predict and indicate drought conditions in China [37]. The latent heat fluxes (LHF), U and V winds and convective precipitation rates were obtained from National Centers for Environmental Prediction/National Center for At-

mospheric Research (NCEP/NCAR) reanalysis data (https://psl.noaa.gov/data/gridded/ (accessed on 5 September 2022)).

*2.2. Methods*

Recently, various machine learning (ML) algorithms and artificial intelligence such as logistic regression (LR), decision trees, ANN (artificial neural networks) random forests, and SVM (support vector machines) have become hugely important due to their large data handling capacities and remarkable accuracy level [38]. Appropriate methods and models have a crucial role in drought monitoring and management. The Standardized Precipitation Evapotranspiration Index (SPEI) and Standardized Soil Moisture Index (SSMI) are used as drought indices for monitoring meteorological drought and soil drought.

2.2.1. Drought Index

Standardized Precipitation Evapotranspiration Index (SPEI)

SPEI was proposed by Vicente-Serrano et al., and Santiago Beguería et al. [24] reconsidered the parameter-fitting effect of SPEI and the influence of the evapotranspiration model. SPEI represents the combined effect of changes in precipitation (*P*) and potential evapotranspiration (*PET*) on drought; and *PET* is calculated using the Penman–Monteith evaporation formula. Vicente-Serrano [21] demonstrated that using the log-logistic distribution to standardize the difference (*D*) between precipitation and evaporation to calculate the SPEI yields optimal results.

$$D = P - PET \tag{1}$$

Next, the probability function distribution is fit to *D* using the log-logistic distribution:

$$F(D) = \left[ 1 + \left( \frac{\alpha}{D - \gamma} \right)^{\beta} \right]^{-1} \tag{2}$$

where $\alpha$, $\beta$, and $\gamma$ are the scale, shape, and position parameters, respectively. These parameters are estimated by probability-weighted distance methods (PWMs) based on empirical frequencies:

$$w_s = \frac{1}{N} \sum_{i=1}^{N} (1 - F_i)^s D_i \tag{3}$$

where $w_s$ is the s-order PWM, $N$ is the number of data points, and $F_i$ is the frequency estimate. Using PWMs to estimate the parameters, the formula used to calculate SPEI is expressed as follows:

$$SPEI = W - \frac{C_0 + C_2 + C_3 W^2}{1 + d_1 W + d_2 W^2 + d_3 W^3}$$
$$W = -2ln(P) \tag{4}$$

In the above equation, when $p \leq 0.5$, $p$ is the cumulative probability, and $p = 1 - F_{(x)}$; when $p > 0.5$, then $p = 1 - p$. The other parameters were set as follows: $C_0 = 2.515517$, $C_1 = 0.802853$, $C_2 = 0.010328$, $d_1 = 1.432788$, $d_2 = 0.189269$ and $d_3 = 0.001308$.

Standardized Soil Moisture Index (SSMI)

The SSMI is used as an index to evaluate agricultural drought by standardizing soil moisture information. The process by which soil moisture is normalized requires determining the probability distribution of a soil moisture data series. The probability distribution functions commonly used for drought indices mainly include the gamma distribution, Pearson III distribution, empirical cumulative probability distribution, normal distribution and log-logistic distribution [34]. After applying the Kolmogorov–Smirnov

(K-S) test, soil moisture data conform to a normal distribution. Therefore, the method used to calculate the SSMI can be expressed as follows:

$$SSMI = \frac{SM - \overline{SM}}{\sigma}$$ (5)

where *SM* is the soil moisture value at a certain time and a certain location, $\overline{SM}$ is the multiyear average soil moisture value at a certain location, and $\sigma$ is the multiyear soil moisture standard deviation at a certain location. When the SSMI value is negative, the soil moisture is below the normal value, meaning that soil moisture is deficient; when the SSMI value is positive, it means that the soil moisture is in surplus. Referring to the drought classification system proposed by Mckee et al. [21] and the World Meteorological Organization (WMO), the SSMI and SPEI are divided into Table 1.

**Table 1.** Classification schemes of the SPEI and SSMI.

| SPEI/SSMI | Class | SPEI/SSMI | Class |
|---|---|---|---|
| >2.00 | Extremely wet | −0.99–0 | Mild drought |
| 1.99–1.50 | Damp | −1.49––1 | Moderate drought |
| 1.49–1.00 | Moist | −1.99––1.5 | Severe drought |
| 0.99–0 | Normal | <−2.00 | Extreme drought |

2.2.2. Statistical Analysis
Mann–Kendall (M-K) Analysis

The Mann–Kendall (M–K) test is an effective tool used to extract sequential variation trends; this method is widely used in analyses of climate parameters and hydrological sequences [34]. The M–K method is famous for its wide application range, low artificiality and high quantification degree. In this study, we used the M–K test method to study the time series trend changes in the SPEI and the linear trend changes in the temporal coefficient of the principal components. The following equation is used:

$$S = \sum_{i=1}^{n-1} \sum_{j=i+1}^{n} sgn(X_i - X_j)$$ (6)

where $X_j$ is the $j_{th}$ data value of the time series, *n* is the length of the data sample and *sgn* is the sign function. The sign function can be defined as follows:

$$sgn(\theta) = \begin{cases} 1 & (\theta > 0) \\ 0 & (\theta = 0) \\ -1 & (\theta < 0) \end{cases}$$ (7)

In the above formula, when $X_i - X_j$, is less than, equal to or greater than zero, the values −1, 0 and 1, respectively, are assigned; when the M–K statistic *S* is greater than, equal to, or less than zero, the following equations are applied, respectively:

$$Z = \begin{cases} (S-1)/\sqrt{n(n-1)(2n+5)/18} & S > 0 \\ 0 & S = 0 \\ (S+1)/\sqrt{n(n-1)(2n+5)/18} & S < 0 \end{cases}$$ (8)

A positive *Z* value indicates an increasing trend, while a negative value indicates a decreasing trend. When the Z value is greater than or equal to 1.96, a significant upward trend exists, and vice versa.

Importance of Resampling Tests

We constructed split violin plots showing the value ranges corresponding to the monthly-scale climate indices (Nino 3.4, PDO and AMO) and latent heat in months that suffered from drought or non-drought conditions. To determine whether each climate index differed significantly between the dry and non-drought years in each case, resampling tests were performed. The number of dry and non-drought years was determined for each case; then, similar values were randomly selected from all climate index time series using replacement sampling. This process was repeated 1000 times, and the original arid and non-arid climate index anomalies were compared with those obtained from these 1000 randomly sampled cases [35].

## 3. Results

### 3.1. Spatiotemporal Characteristic of Climate Factors and Drought Indices

The average of SPEI (Figure 1a) shows a regionalized distribution, with negative values (representing drought conditions) mainly distributed in the Liaohe River Basin, on the North China Plain, on the Loess Plateau and in the five southwestern provinces, with a minimum value of −0.3; positive values are mainly concentrated on the Qinghai-Tibet Plateau, in the northwest arid region and in the Heilongjiang River Basin, with a maxi-mum value of 0.3. The latent heat distribution (Figure 1b) is mainly concentrated with obvious climatic-zoning characteristics. Average LHF values can be found in the transition area of the monsoon climate zone, while the extreme value also appears in this transition zone, with a value of 250 w/m$^2$; the average LHF south of this transition area can reach 125 w/m$^2$, the average value to the north is less than 125 w/m$^2$, and the average value to the northeast is close to 0 w/m$^2$. The average soil moisture distribution (Figure 1c) shows that areas with high soil moisture are concentrated in the northeast and south regions of the Yangtze River; the average SPEI distribution shows that meteorological drought conditions have gradually increased in the Liaohe River Basin, on the North China Plain, and on the Loess Plateau, while the average soil moisture value has been maintained only at 25%; thus, the impacts of drought in these areas are very serious.

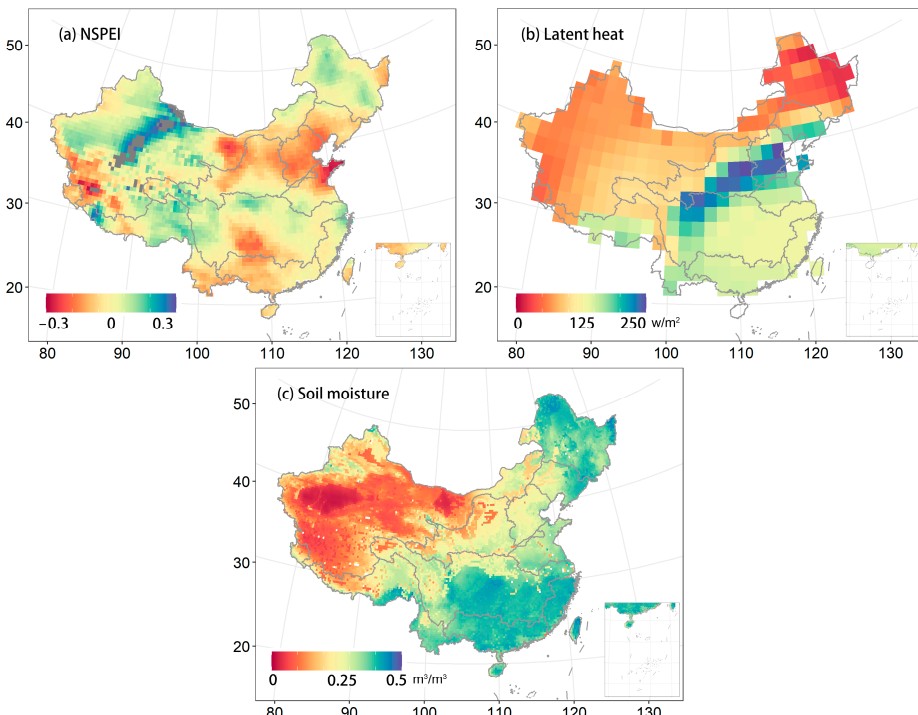

**Figure 1.** Spatial characteristics of annual SPEI, LHF and soil moisture.

By analyzing the spatial distribution of drought indicators in this paper, we synthesized the AMO, PDO and Nino3.4 data annually (Figure 2) and analyzed the temporal changes in these climatic factors. The change rule of the annual average AMO (Figure 2a) index was found to remain constant at −0.1 from 1980–1990, began to increase rapidly from approximately 1990–2000, and remained at approximately 0.2 after 2000. The annual average PDO series (Figure 2b) showed a regular pattern of first falling and then rising, with the PDO falling from 1 to −1 (with the lowest point occurring in 2010) and then rising to 1 in approximately 2018. Compared to the trends of the other two analyzed climate factors, the Nino3.4 trend was relatively gentle (Figure 2c). AMO changed dramatically from 1980 to 2018, rising from −0.12 to 0.13 (Table 2). PDO did not change much according to the results, and Nino3.4 showed a similar upward trend to AMO from −0.02 to 0.25.

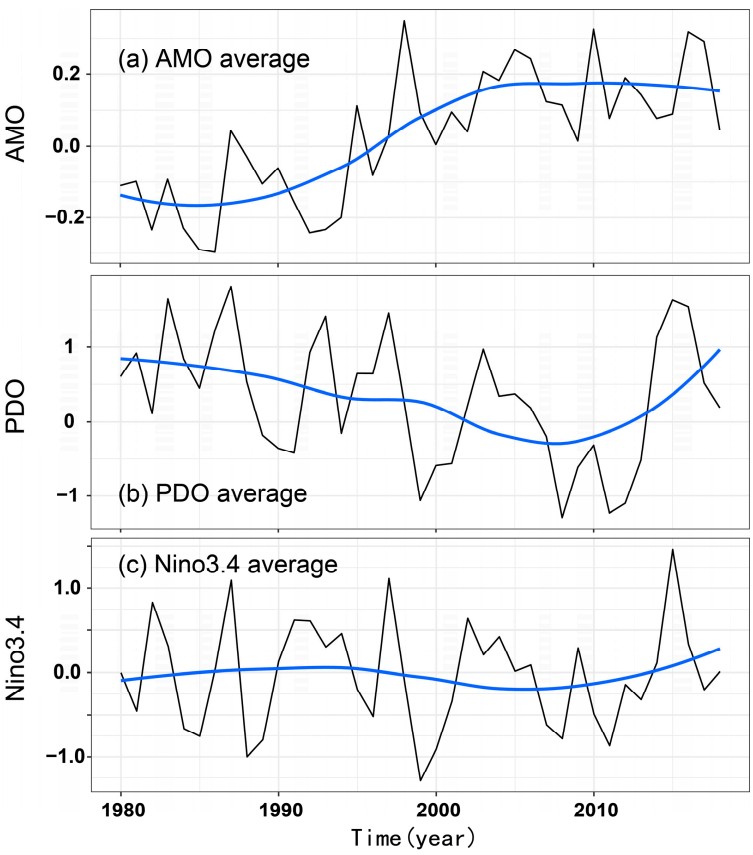

**Figure 2.** Temporal characteristics of annual average in AMO, PDO and Nino3.4 (The black line is the annual evaluation and the blue line is the fitting line for the losses function).

**Table 2.** 1980–2018 Annual average change in AMO, PDO and Nino3.4.

| Climatic factors | 1980 | 2018 |
| :---: | :---: | :---: |
| AMO | −0.12 | 0.13 |
| PDO | 0.90 | 1.00 |
| Nino3.4 | −0.02 | 0.25 |

### 3.2. Soil and Meteorological Drought Temporal Response to Climatic Factors in China

The average distributions of the LHF, PDO, AMO and Nino3.4 indices in dry months and non-drought months in China were analyzed across the whole region of China (Figure 3). SPEI-1 (on a 1-month scale) (Figure 3a1) showed small differences between drought and non-drought months and LHF distribution, and it can be said that the effect of LHF on short-term meteorological drought is relatively small (difference in means < 0.1). For SPEI-12 (Figure 3b1), the mean value of the LHF distribution was higher in dry months

(mean value 0.7) than in non-dry months (mean value 0.5), indicating that the increase in latent heat mitigated long-term drought conditions. Soil drought (Figure 3c1) reflects more obvious changes in the LHF in dry months than in non-drought months, and negative latent heat values are more concentrated in non-drought months. From the above analysis, it can be seen that latent heat can influence the change of long-term drought, and the effect on SPEI-12 is greater than that of SPEI-1.

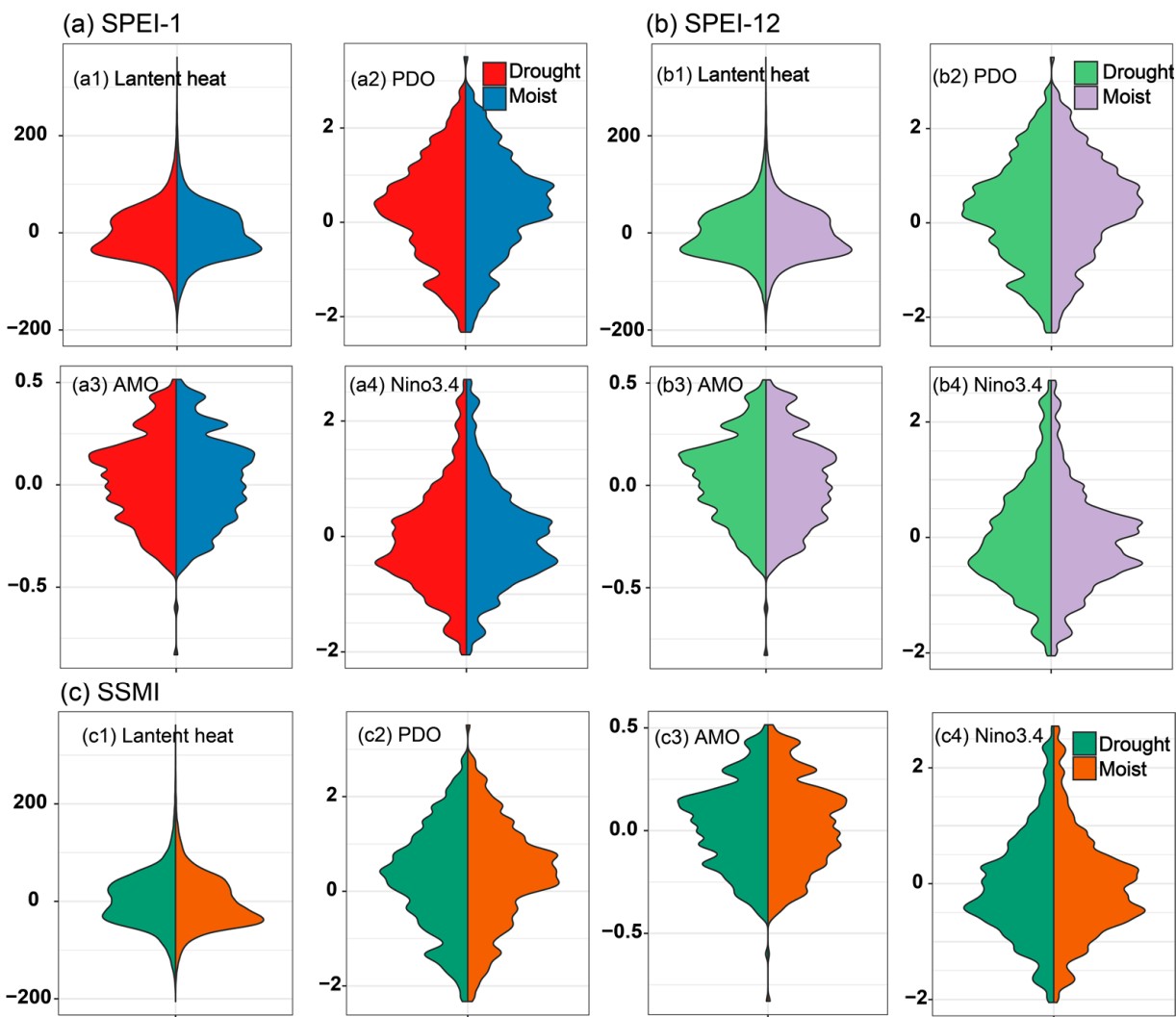

**Figure 3.** Response of LHF, PDO, AMO, and Nino3.4 to dry and non-drought months indices in China.

The PDO distributions of all drought indices were essentially similar in drought and non-drought months (Figure 3a2,b2,c2) (mean difference < 0.2). The PDO values in drought months tended to be negative, indicating that the probability of drought in China increases when the PDO is negative. AMO values were higher in drought months than in non-drought months, but the difference in SPEI-1 (mean value less than 0.1) (Figure a3) was less pronounced than in SPEI-12 (Figure b3) or soil drought (Figure c3). Short-term meteorological droughts have less of an impact on the entire Chinese region than other drought types. The data distribution of the Nino3.4 index showed the largest difference between dry months and non-drought months among the analyzed indices. The average Nino3.4 index values in dry months were mainly concentrated at approximately −0.5, while in non-drought months, two peaks at −0.5 and 0.5 were observed. This finding shows that when the Nino3.4 index is negative, the drought conditions in mainland China are closely related to the occurrence of a La Niña event.

The SPEI-1 (1-month scale), SPEI-12 (12-month scale) and SSMI values were compared among the ten major river basins in China, and these series were assessed with regard to periods of different types of drought and non-drought periods by analyzing the distributions of the climate indices. The latent heat data distributions differed extensively among different watersheds (Figure 4a). The LHF in the Haihe River, Huaihe River, Yellow River and Liaohe River basins exhibited strip-like distributions, mainly because these areas are concentrated in the LHF transition regions (Figure 1b). The SPEI-1 (Figure 4a1) reflected little difference in the LHF distributions between dry months and non-drought months, consistent with the above results, while for SPEI-12 and soil moisture, the average of values in the Heilongjiang River Basin, Haihe River Basin, Huaihe River Basin, and Liaohe River Basin of the Yellow River basin in dry months was significantly lower than those in non-drought months (mean difference of 0.3). In addition, the relatively low LHF values and the relatively high drought probability values indicated that these droughts are caused by insufficient precipitation. The average LHF values in the inflow area, the southwest outflow area and the Pearl River Basin were higher in dry months than in non-drought months (mean value above 0.2), indicating that the drought events in these watersheds were caused mainly by extensive water evaporation.

The PDO distributions derived in the ten major watersheds were relatively similar, and no large differences were found among the watersheds like those seen for the LHF; however, different watersheds showed different data distributions. The SPEI-1 and SPEI-12 distributions indicate that in the Heilongjiang River Basin, Southwest Outflow Area and Pearl River Basin, the average PDO is greater in dry months than in non-arid months. When the PDO value is positive, the probability of drought increases in these basins. The mean dry month values in the Haihe River, Yellow River, Huaihe River, and Liaohe River basins were smaller than those in non-drought months, indicating that these indices were negatively correlated with the PDO in these basins. The relationship between soil drought and the PDO index was quite different from that between meteorological drought and the PDO index. This difference manifested in the Haihe River Basin, Heilongjiang Basin, Huaihe River Basin and Yellow River Basin. This relationship exhibited more average values, and the observed differences may be related to irrigation activities [28]. The AMO index distribution was similar to that of the SPEI-1 in both dry months and non-drought months, while the SPEI-12 and soil dryness results were also similar, for which the average values found in the Heilongjiang River Basin, the Yellow River Basin and the Yangtze River Basin in dry months were higher than those values in non-drought months, indicating that the effect of the AMO on short-term drought is smaller than the effects on long-term and agricultural drought. Analyzing the relationship between the Nino3.4 index and the various watersheds, on the whole, the results obtained for the northwest inner watershed and other watersheds were the most significant, and the dry month values were smaller than the values in non-arid months; the Pearl River Basin and the southeast coastal watershed showed the opposite characteristics. As a result, the dry month values were larger than the non-drought month values, and the Nino3.4 index reflected the relatively great impacts of drought in these two watersheds.

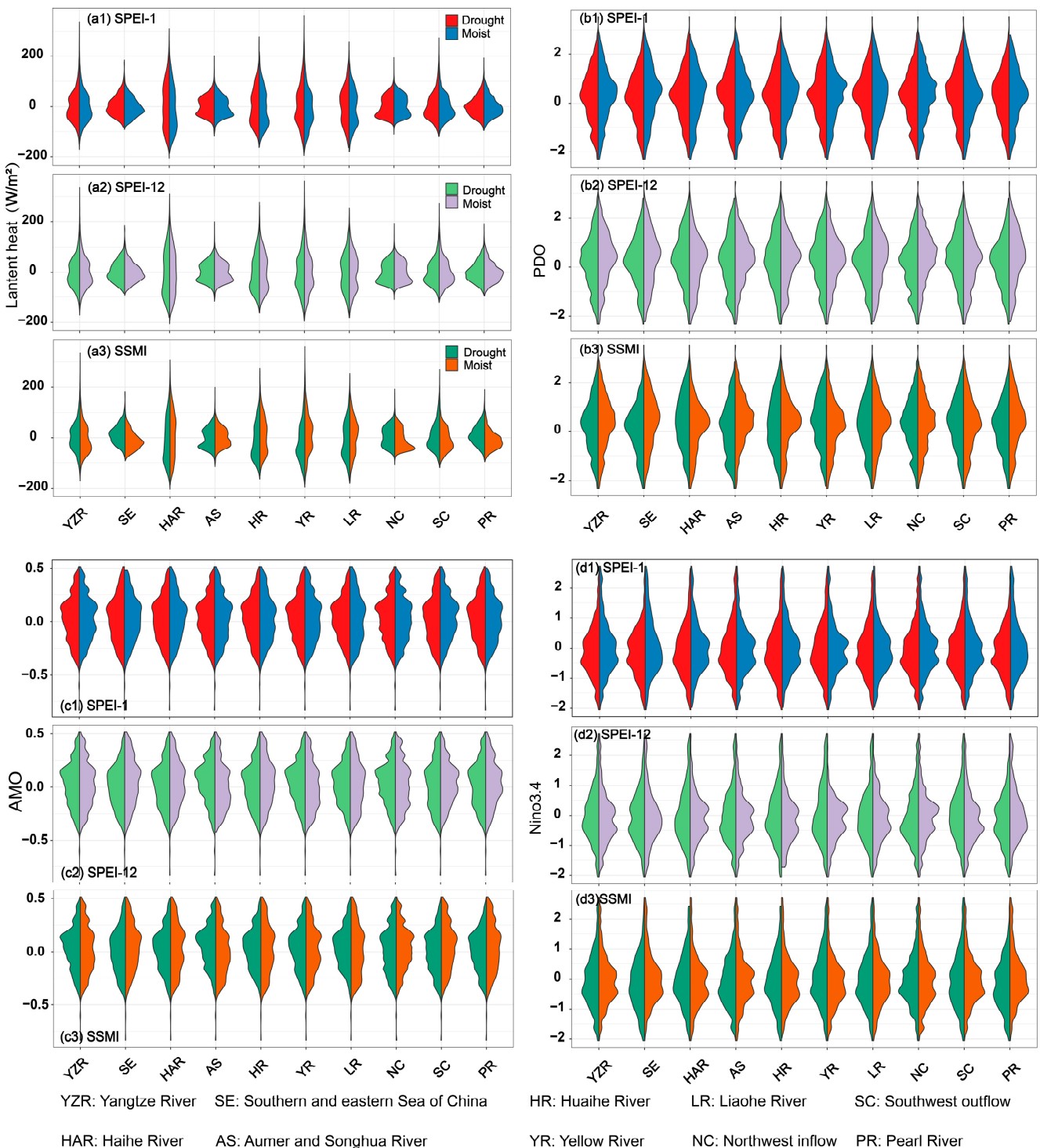

**Figure 4.** Response of LHF, PDO, AMO and Nino3.4 to the dry and non-drought months of in the major watersheds of China.

### 3.3. Soil and Meteorological Drought Spatial Response to Climatic Factors in China

To analyze the spatial response of soil and meteorological drought to climate indices, this study used the spatial distribution of correlations. Figure 5 shows the distributions of the spatial correlations between SPEI-1 and the LHF, PDO, AMO and Nino3.4 indices. The SPEI-1 and LHF relation showed a significant regional distribution (Figure 5a); regions with negative correlations were mainly concentrated in the monsoon climate region, where the minimum value of the correlation coefficient was $-0.20$, and the regions with the smallest

values were concentrated in the Yellow River Basin and Haihe River Basin; this finding indicates that the short-term meteorological drought events in these two regions were caused by a lack of precipitation. The northwest inner flow area exhibited a significant positive correlation between the SPEI-1 and the LHF. Regions in which significant positive correlations were found between the SPEI-1 and PDO index (Figure 5b) were located in China's southeast coastal watershed, the Pearl River watershed, the northwest inner watershed, and the eastern Heilongjiang watershed. These areas corresponded to significant short-term drought and index changes in the northern Pacific. The maximum value of the correlation coefficient was 0.21. The AMO represents the multidecadal oscillations that occur in the North Atlantic. Figure 5c shows that the short-term meteorological drought events in the northwest inland region were significantly negatively correlated with the AMO (with a correlation coefficient of 0.19), but that this relationship was not significant in other regions. The Nino3.4 index showed a significant positive correlation with short-term drought on the southeastern coast of China, with the highest correlation coefficient reaching 0.23.

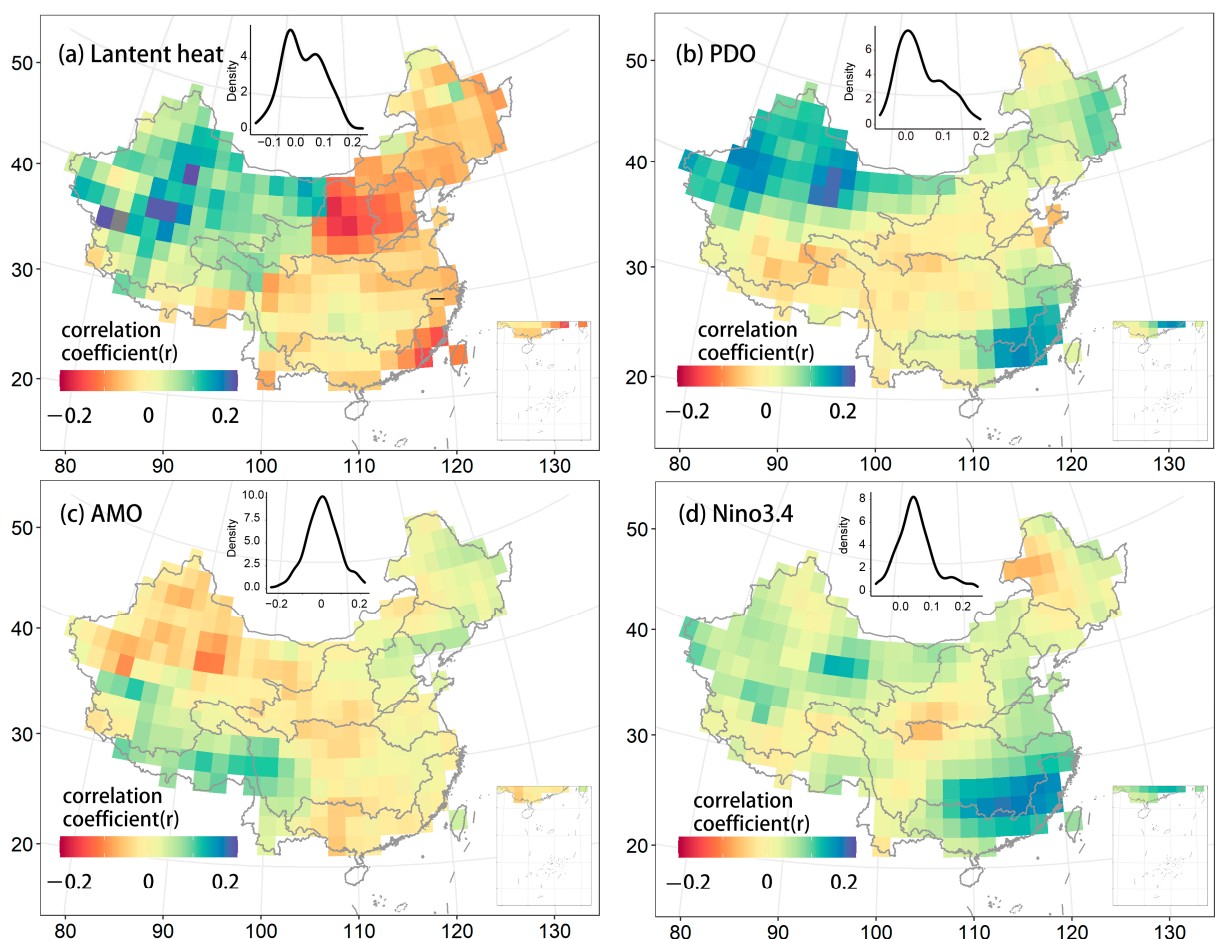

**Figure 5.** Spatial distribution of SPEI-1 response to LHF, PDO, AMO and Nino3.4.

When analyzing the relationship between meteorological drought and a climate index at only a single scale, an intrinsic relationship often cannot be found; thus, in this paper, we analyzed the spatial relationships between the SPEI-12 and the LHF, PDO, AMO and Nino3.4 indices (Figure 6). The spatial distribution of the correlation between the latent heat flux and SPEI-12 was quite different from the spatial distribution of the SPEI-1 index itself. The areas where the SPEI-12 and latent heat flux were negatively correlated are mainly concentrated in the southeastern coastal watershed, the Pearl River watershed and the middle and lower reaches of the Yangtze River. The correlation coefficient in these regions

reached −0.19, suggesting that long-term meteorological droughts are highly susceptible to precipitation shortages in these regions (Figure 6a). The spatial distribution of the correlation between the SPEI-12 and PDO index was similar to that of the correlation between the SPEI-1 and PDO index (Figure 6b), and areas with significant positive correlations grew; regions with significant positive correlations are concentrated in the Pearl River Basin and the Southeast Coastal Basin, most of the Yangtze River Basin, in the southern part of the Huaihe River Basin, and in the entire Northwest Inner Flow Region; in these areas, the correlation coefficient increased to 0.3, while negative correlations were identified in the transition zone between arid and semiarid regions. The impacts of this climatic factor were relatively deep and wide, and this finding is also consistent with the results analyzed in Figure 4. The spatial distribution of the correlation coefficients between the AMO index and SPEI-12 was consistent with that of the relation between the AMO index and SPEI-1 (Figure 6c). Negative-correlation areas are mainly concentrated in the northwest inflow area. The impact of the AMO index on long-term meteorological drought is thus relatively widespread and severe. The correlation coefficient of −0.1 decreased to −0.18 in some regions. The distribution law of the correlation coefficients derived between the Nino3.4 index and SPEI-12 was the same as that of the correlation coefficients obtained between the Nino3.4 index and SPEI-1 (Figure 6b), except the SPEI-12 values in Yellow River Basin and the northwest of the Yangtze River Basin exhibited significant negative correlations with Nino3.4; in addition, the center of the significant positive-correlation range expands from the southeast coastal watershed to include to the Pearl River Basin. From the above analysis, it can be seen that the effects of Nino3.4 on long-term meteorological drought in China are mainly manifested in central China.

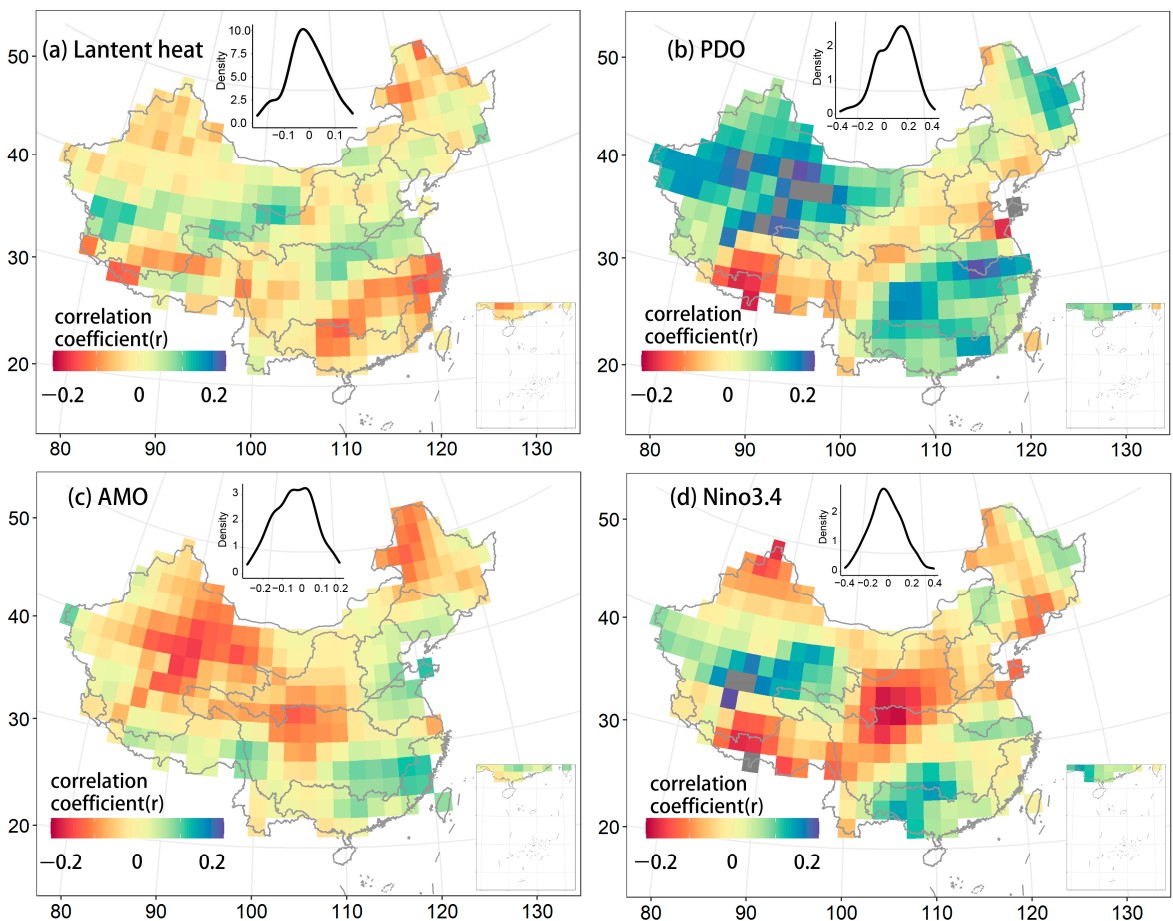

**Figure 6.** Spatial distribution of SPEI-12 response to LHF, PDO, AMO and Nino3.4.

Explorations of the spatial relationships between meteorological drought and climatic factors often cannot fully reflect the impacts of climatic factors on drought in China. Therefore, analyzing the relationships between soil drought and climatic factors can more comprehensively reveal the evolutionary mechanisms of drought in China. The spatial distribution of the correlation between the SSMI and LHF is similar to that found for SPEI-12 (Figure 7a). The negative-correlation areas are concentrated in the Pearl River Basin, along the southeast coast, in the middle and lower reaches of the Yangtze River, and in the northern part of the northwest inner flow area; in these regions, the correlation coefficient reaches −0.8. The areas exhibiting correlation are consistent with the negative-correlation areas obtained with regard to SPEI-1, and the maximum correlation coefficient is 0.78. This phenomenon reflects the fact that short-term drought and long-term drought have opposing effects on water evaporation; the LHF affects both long-term meteorological drought and soil drought under the same rules. The correlation distributions derived between the PDO and the three drought indices are essentially similar (Figure 7b), but the correlation between soil drought and PDO is the strongest, with correlation coefficients reaching 0.4, suggesting a strong correlation. The correlation distributions of the AMO index, Nino3.4 index and SSMI are similar to that of SPEI-12.

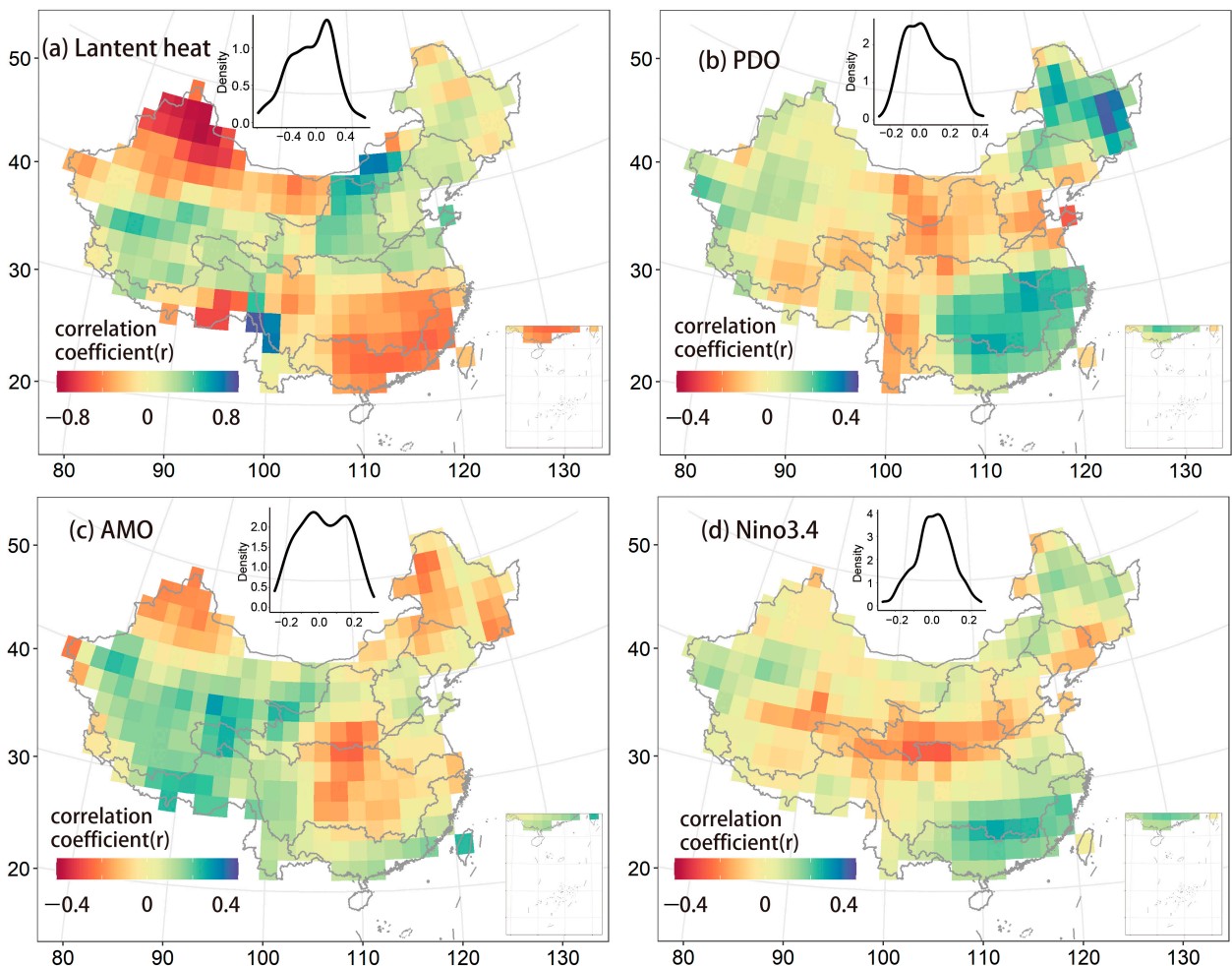

**Figure 7.** Spatial distribution of SSMI response to LHF, PDO, AMO and Nino3.4.

## 4. Discussion

The causes of soil and meteorological drought are complex and closely related to meteorological conditions and climatic factors [5,8,12,18,36]. Therefore, to evaluate the internal relationships between drought and climatic factors, it is necessary to perform a comprehensive evaluation from different aspects [12,13,33]. In past evaluations of the

relationship between soil and meteorological drought and climatic factors, assessments were often carried out by analyzing the correlations between time series and discussing the physical causes of drought, while the spatial distributions of these correlations and the differences between dry and wet months were not considered [33].

In general, the northeastern parts of China had more severe drought conditions than the southern parts [37]. However, the humid climate zone in the south also experienced severe drought conditions, though not as much as for northern parts of China [39]. In contrast, the seasonal variation in latent heat flux is largest in Southeast China and smallest in Northwest China [40].

Precipitation distribution on the Tibetan Plateau is clearly influenced by topography, which gradually decreases from southeast to northwest. Precipitation is mainly concentrated in the southern foothills of the Himalayas [41]. Several studies have shown that the diurnal variation of rainfall undergoes a clear seasonal and sub-seasonal evolution. During the cold season, the regional contrast of diurnal peaks in rainfall decreases, with the greatest amount of rainfall occurring in the early morning in most of southern China [42]. Additionally, there is a decreasing trend in convective precipitation [43]. Thus, drought is further enhanced by climate factors. However, there are still some shortcomings in our study. With the increased number of meteorological data products available, the spatial resolution of available data will increase, the accuracy will increase, and the accuracy of the research results obtained using these products will also improve.

When discussing the differences between dry months and non-drought months among different basins, it is necessary to further consider irrigation and other artificial factors, especially in the Huaihe River Basin, the Yellow River Basin, and the Haihe River Basin, where many artificial irrigation factors are present, leading to the particularities of the abovementioned areas with regard to the differences between soil drought and long-term meteorological drought. SSMI information is used in this paper to characterize soil moisture, and this index contains some information on artificial irrigation.

This study attempted to reveal the causes of soil and meteorological drought, but the research method is not deep enough. Thus, this study can be refined in the future by using geological methods, such as random forest machine learning, geographically weighted regression models, or geographic detectors, etc. The formation mechanisms of drought in China need to be further explored in the future.

## 5. Conclusions

Focusing on the response of regional drought to climate factors is an important indicator for studying the impact of global climate change on drought. At the global scale, the local response to atmospheric circulation allows some regional-scale patterns to be identified. In this study, the soil and meteorological drought response to climate factors was explored, along with the change in response at different spatial and temporal scales. We then finally attempted to explain the results in terms of changes in circulation. Specific research conclusions are described as follows:

(1) Meteorological drought has gradually increased in the Liaohe River Basin, in the North China Plain, and on the Loess Plateau, while average soil moisture has been maintained at only approximately 25%. The impacts of drought are very serious in these regions. These results are consistent with previous studies.

(2) The temporal response (3-month scale) of LHF to short-term soil and meteorological drought was chronically higher in dry months than in non-dry months. Short-term soil and meteorological drought and LHF showed a clear regional distribution, and the negatively correlated regions were mainly concentrated in monsoon climate zones. The regions with negative responses to long-term soil (12-month scale) and meteorological drought and LHF were mainly concentrated in the southeastern coastal basin. Long-term soil (12-month scale) and meteorological drought are more vulnerable to precipitation shortage than other types of drought.

(3)    Short-term and long-term soil and meteorological drought have a highly responsive significant relationship with the PDO, exhibiting changes in the southeastern coastal watersheds, the Pearl River watershed, the northwestern inner watershed and the eastern part of the Heilongjiang watershed, with a maximum correlation coefficient is 0.21 ($p < 0.01$). The short-term soil and meteorological drought conditions in the northwest inland region showed significant negative correlations with the AMO (with a correlation coefficient of $-0.19$, $p < 0.01$), but this relationship was not significant in other regions. The Nino3.4 index was significantly positively correlated with short-term soil and meteorological drought along the southeastern coast of China, with the highest correlation coefficient reaching 0.23 ($p < 0.01$). A decrease in the rate of convective precipitation leads to strengthened links between soil and meteorological drought and climatic factors.

Overall, this study helps to uncover the patterns of drought changes under the combined effects of meteorological and soil drought, and can also be used globally to identify the local development patterns of drought under climate change.

**Author Contributions:** Conceptualization: Y.L.; Data curation: L.L.; Formal analysis: L.L. and L.Z.; Methodology: L.L. and Y.L.; Supervision: Y.L. and L.Z.; Validation: L.L., L.Z. and Y.L.; Writing—original draft: L.L.; Writing—Reviewing and Editing: L.Z. and Y.L.; Funding acquisition: Y.L. and L.L. All authors have read and agreed to the published version of the manuscript.

**Funding:** This work was supported by the National Natural Science Foundation of China (grant no. 52179015), and the Key Technologies R & D and Promotion program of Henan (202102110128).

**Data Availability Statement:** Climatic Research Unit (CRU) 4.0 meteorological data (temperature, precipitation, humidity, wind speed, etc.) obtained at a spatial resolution of $0.5° \times 0.5°$ with a time series length of 1901–2018 (https://lr1.uea.ac.uk/cru/data 5 September 2022). The soil moisture data were collected from the Global Land Evaporation Amsterdam Model (GLEAM) (https://www.gleam.eu/ 5 September 2022) with a time span of 1980–2018 and a spatial resolution of $0.25° \times 0.25°$. The Pacific Decadal Oscillation (PDO), Atlantic Multidecadal Oscillation (AMO), and Nino3.4 (East Central Tropical Pacific SST) climate factors were obtained from the National Oceanic and Atmospheric Administration (NOAA) Physical Sciences Laboratory of the National Weather Service (https://psl.noaa.gov/data/climateindices./list/ 5 September 2022). The latent heat fluxes (LHF), U and V winds and convective precipitation rates were obtained from National Centers for Environmental Prediction/National Center for Atmospheric Research (NCEP/NCAR) re-analysis data (https://psl.noaa.gov/data/gridded/ 5 September 2022).

**Conflicts of Interest:** The authors declared that they have no conflicts of interest to this work.

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
