# Peer review of "Spatiotemporal Characteristics of Meteorological and Agricultural Droughts in China: Change Patterns and Causes"

_agriculture, doi:10.3390/agriculture13020265_

Round 1

Reviewer 1 Report

Manuscript Ref No.: agriculture-2136029

The manuscript has some shortcomings which need to be improved prior to its publication. My recommendation is that the article needs Major Revisions before it can be considered for publication.

1.     Abstract: The abstract is a bit generic. Please add some more information regarding your results. It should be improved in a quantitative way.

2.     Author should specify the key objectives of this research work in last paragraph of introduction section.

3.     Introduction is generalized. I would recommend following recent research articles to reconstruct this section with extensive literature.

Climate change and groundwater overdraft impacts on agricultural drought in India: Vulnerability assessment, food security measures and policy recommendation”.

 Drought risk assessment: integrating meteorological, hydrological, agricultural and socio-economic factors using ensemble models and geospatial techniques

4.     Methodology section is weakly written. So, my suggestion is to reconstruct it. Author need to review more about adopted methods and machine learning algorithms.

Characterization of groundwater potential zones in water-scarce hardrock regions using data driven model

5.     Discussion section should be written by comparing with already published articles in this concept.

6.     In conclusion section, you have to mention the implications of your research and how it makes a footprint in scientific research. Try to incorporate your work to global interest how this research has worldwide importance. It will be interesting for the readers.

7.     Reference: Re-check the whole reference just to make sure you have added all the references that you cited in your manuscript.

8.     Apart from this the quality of the overall paper is very good. I prefer this article with acceptable with major modifications.

Author Response

  1. Abstract: The abstract is a bit generic. Please add some more information regarding your results. It should be improved in a quantitative way.

Reply: Thanks for the constructive comment. We have improved the abstract in a quantitative way in Lines 8-22.

  1. Author should specify the key objectives of this research work in last paragraph of introduction section.

Reply: Thanks for the constructive comment. We have rewritten the objectives in Lines 68-72.

“Considering these research gaps, the objectives of this study are: (1) Exploring temporal and spatial correlation coefficients between climatic factors and SMD in dry and non-dry months; (2) to explore the response of drought on different time scales to climate factors. These insights provide a reference for drought preparedness and shaping policy recommendations for agricultural and industrial sectors.”

  1. Introduction is generalized. I would recommend following recent research articles to reconstruct this section with extensive literature.

Climate change and groundwater overdraft impacts on agricultural drought in India: Vulnerability assessment, food security measures and policy recommendation”.

 Drought risk assessment: integrating meteorological, hydrological, agricultural and socio-economic factors using ensemble models and geospatial techniques

Reply: Thank you very much for your helpful suggestions and interesting references. We have reorganized the introduction and added essential references (see Lines 47-53).

  1. Methodology section is weakly written. So, my suggestion is to reconstruct it. Author need to review more about adopted methods and machine learning algorithms.

“Characterization of groundwater potential zones in water-scarce hardrock regions using data driven model”

Reply: Thanks for the constructive comment. We have re-structured the methodology and added a review of machine learning algorithms in Lines 89-166.

  1. Discussion section should be written by comparing with already published articles in this concept.

Reply: Thanks for the constructive comment. We have rewritten the discussion section, also removed Figure 8 and rewritten it by comparing published articles in Lines 355-390.

  1. In conclusion section, you have to mention the implications of your research and how it makes a footprint in scientific research. Try to incorporate your work to global interest how this research has worldwide importance. It will be interesting for the readers.

Reply: Thanks for the constructive comment. In the concluding section, we have rewritten the implications of the study and the work of integrating this study with global interests in Lines 393-399 and Lines 423-426.

“Focusing on the response of regional drought to climate factors is an important indicator for studying the impact of global climate change on drought. At the global scale, the local response to atmospheric circulation allows some regional scale patterns to be identified.”

“Overall, this study helps to uncover the patterns of drought changes under the combined effects of meteorological and soil drought, and can also be used globally to identify the local development patterns of drought under climate change.”

  1. Reference: Re-check the whole reference just to make sure you have added all the references that you cited in your manuscript.

Reply: Thanks for the constructive comment. We have re-checked the whole reference.

  1. Apart from this the quality of the overall paper is very good. I prefer this article with acceptable with major modifications.

Reply: Thank you very much for your approval of our paper.

Reviewer 2 Report

Thank you very much for inviting me to review this manuscript.

I read your manuscript "Spatiotemporal characteristics of meteorological and agricultural droughts in China: change patterns and causes" with great interest and a well-written manuscript.

Minor revisions are all that is required to make this manuscript more complete.

Table 1: Extreme - Were the authors referring to extreme dry? If this is the case, please revise it.

The authors should explain why they selected the Nino3.4 index and Webpages data for this study.

Author Response

We would like to thank Reviewer #2 most sincerely for his/her constructive comments and valuable suggestions, which have helped us to thoroughly revise and improve the manuscript. Below please find our responses to Reviewer #2’s comments, and we have also incorporated them to the revised manuscript (RM).

  1. Table 1: Extreme - Were the authors referring to extreme dry? If this is the case, please revise it.

Reply: We have revised it in Table 1.

  1. The authors should explain why they selected the Nino3.4 index and Webpages data for this study.

Reply: Thanks for the constructive comment. Previous study has shown that NIno3.4 is used to predict and indicate drought conditions in China, therefore we choose Nino3.4. We have added the explanation to the manuscript in Lines 80-85.

“The Pacific Decadal Oscillation (PDO), Atlantic Multidecadal Oscillation (AMO), and Nino3.4 (East Central Tropical Pacific SST) climate factors were obtained from the Nation-al Oceanic and Atmospheric Administration (NOAA) Physical Sciences Laboratory of the National Weather Service (https://psl.noaa.gov/data/climateindices./list/). It has been shown that the Nino3.4 is used to predict and indicate drought conditions in China [35].”

Zhang, Y.; Hao, Z.C.; Feng, S.F.; Zhang, X.; Xu, Y.; Hao, F.H. Agricultural drought prediction in China based on drought propagation and large-scale drivers. Agr Water Manage 2021, 255, doi:ARTN 10702810.1016/j.agwat.2021.107028.

Reviewer 3 Report

The study presents interesting relationships between the occurrence of drought in China and selected climatic indicators. However, I suggest omitting Figure 8, which does not quite fit the topic of the study. However, the results should be supplemented with the temporal variability of drought indices (SPEI and SSMI), as well as LHF and soil moisture in China. This will show the changes that have taken place in the last 40 years in the values of these variables in different regions of the country.

In the introduction or conclusions there should be a reference to the relationship between the occurrence of drought in China and changes in air temperature in this country.

Line 153 - It should be Fig. 1c

Author Response

We would like to thank Reviewer #3 most sincerely for his/her constructive comments and valuable suggestions, which have helped us to thoroughly revise and improve the manuscript. Below please find our responses to Reviewer #3’s comments, and we have also incorporated them to the revised manuscript (RM).

  1. However, I suggest omitting Figure 8, which does not quite fit the topic of the study. However, the results should be supplemented with the temporal variability of drought indices (SPEI and SSMI), as well as LHF and soil moisture in China. This will show the changes that have taken place in the last 40 years in the values of these variables in different regions of the country.

Reply: Thanks for the constructive comment. We have removed Figure 8. Since many people did spatial trend changes in SPEI and SSMI, we have rewritten it by comparing published articles in Lines 355-390.

  1. In the introduction or conclusions there should be a reference to the relationship between the occurrence of drought in China and changes in air temperature in this country.

Reply: Thanks for the constructive comment. We have added a description of the relationship between temperature and drought in China in Lines 40-45.

“Some research showed the negative connection between temperature and droughts encom-passes most of China, indicating that high temperatures and droughts are generally con-current in these regions [9]. Drought was most widespread and sustained in the late 1990s and early 2000s, primarily in the southern Yellow River Basin and the northern Yangtze River Basin. [10]. precipitation and potential evapotranspiration are projected to increase throughout China [11]”

  1. Line 153 - It should be Fig. 1c

Reply: Thanks for the constructive comment. We have revised it.

Reviewer 4 Report

1.       The abstract of the article is very generics as not a single numeric value is presented. Results must be presented with the numeric values for the assessment of the threshold of any factor and its deviation.

2.       The sentence starting from the line 25 and ending at line 30 is ambiguous and unable to understanding, especially the end of line 28.

3.       The line 30 “Soil co-occurs with meteorological drought effects on vegetation and 30 the climate” what is co-occurs?

4.       The sentence on the line 33 to 35 is very generic that is very well known that increase in the drought [increase in temperature and decrease in rainfall] will increase the drought.

5.       No need to mention that, its better to present that what is soil drought? How much soil moisture is the critical level for a significant drought that really impacts the agriculture?  

6.       The line 37 is not clear and complete that what these authors want to mention?

7.       Overall, the introduction part of the research article is not at the level of acceptance. No consistency of the work rationale, strong literature review and gaps in the previously reviewed research.

8.       Line 60 is presenting that the authors used “etc” for the data, the parameters of climate used in this study must be mentioned.

9.       Overall, the methodology of the work is well presented and written.

10.   The figures resolution of fig 2(a) to fig 2(c) must be increased as legends are not clears.

11.   It will be better if temporal results must be presented in a table or graphical format.

12.   The line 173 to 178 is methodology of the research work and presented in the results.

13.   The results described in the sections “Soil and Meteorological Drought Temporal Response to Climatic Factors in China” have no numeric presentation, the results are presented with generic terms of higher average values etc.

14.   The line 211 to 221 is same as mentioned above, the pattern of generic writing. Same pattern of writing is continued below as well.

15.   The line 274 to 297, the results are presented in good format where the authors mentioned the level of correlation.

16.   the conclusion must be improved.

Author Response

We would like to thank Reviewer #4 most sincerely for his/her constructive comments and valuable suggestions, which have helped us to thoroughly revise and improve the manuscript. Below please find our responses to Reviewer #4’s comments, and we have also incorporated them to the revised manuscript (RM).

  1. The abstract of the article is very generics as not a single numeric value is presented. Results must be presented with the numeric values for the assessment of the threshold of any factor and its deviation.

Reply: Thanks for the constructive comment. We have improved in a quantitative way in Lines 8-22.

  1. The sentence starting from the line 25 and ending at line 30 is ambiguous and unable to understanding, especially the end of line 28.

Reply: Thanks for the constructive comment. We have revised the sentence in Lines 28-35.

  1. The line 30 “Soil co-occurs with meteorological drought effects on vegetation and 30 the climate” what is co-occurs?

Reply: Thanks for the constructive comment. We have removed the expression "co" and then reinterpreted the definitions of meteorological drought and soil drought separately. Many studies have investigated this simultaneous occurrence. Therefore, it is important to study the simultaneous occurrence of meteorological drought and soil drought in Lines 28-35.

“Meteorological drought usually involves insufficient precipitation with increased atmospheric evaporative demand (e.g., due to high temperatures, high solar radiation, or high winds). Soil drought is considered to have set in when the soil moisture availability to plants has dropped to such a level that it adversely affects the crop yield and hence agricultural profitability [3]. Several studies have shown the climate-catchment-soil control on hydrological droughts and identify key drought drivers (KDD) for drought propagation [4]. Meteorological and agricultural droughts usually combine to affect vegetation [5]. Soil drought and meteorological drought are greatly exacerbated by land–atmosphere feedbacks.”

  1. The sentence on the line 33 to 35 is very generic that is very well known that increase in the drought [increase in temperature and decrease in rainfall] will increase the drought. No need to mention that, its better to present that what is soil drought? How much soil moisture is the critical level for a significant drought that really impacts the agriculture?  

Reply: Thanks for the constructive comment. We have added the explanation about what soil drought is. The critical threshold of severe drought which soil moisture really affects agriculture was also added in the manuscript. (see Lines 30-32 and Lines 47-48).

  1. The line 37 is not clear and complete that what these authors want to mention?

Reply: Thanks for the constructive comment. We have revised this sentence in Lines 56-57.

“There are many indices used for soil and meteorological drought monitoring, such as SPI, SPEI, PDSI and SSMI drought indices [21-27].”

  1. Overall, the introduction part of the research article is not at the level of acceptance. No consistency of the work rationale, strong literature review and gaps in the previously reviewed research.

Reply: Thanks for the constructive comment. We have revised the introduction. We have added an explanation of meteorological drought and soil drought and added literature on the existence of a link between meteorological drought and soil drought. The research objectives have been revised and the entire structure of the preface has been readjusted (see Lines 25-74).

  1. Line 60 is presenting that the authors used “etc” for the data, the parameters of climate used in this study must be mentioned.

Reply: Thanks for the constructive comment. We have added the complete set of climatic parameters used in this study

  1. Overall, the methodology of the work is well presented and written.

Reply: Thank you very much for your approval of our paper.

  1. The figures resolution of fig 2(a) to fig 2(c) must be increased as legends are not clears.

Reply: Thanks for the constructive comment. We have revised the figure. Also we have added the explanation of the legend in the manuscript.

  1. It will be better if temporal results must be presented in a table or graphical format.

Reply: Thanks for the constructive comment. We have added a Table (see Table 2) in and added the description in Lines 195-197.

  1. The line 173 to 178 is methodology of the research work and presented in the results.

Reply: Thanks for the constructive comment. We have moved this part to the methodology.

  1. The results described in the sections “Soil and Meteorological Drought Temporal Response to Climatic Factors in China” have no numeric presentation, the results are presented with generic terms of higher average values etc.

Reply: Thanks for the constructive comment. We have added quantitative descriptions in Lines 171-197.

  1. The line 211 to 221 is same as mentioned above, the pattern of generic writing. Same pattern of writing is continued below as well.

Reply: Thanks for the constructive comment. We have added quantitative descriptions in Lines 207-231.

  1. The line 274 to 297, the results are presented in good format where the authors mentioned the level of correlation.

Reply: Thanks for the constructive comment. We have added the analysis of the significance of the correlation coefficient in Lines 280-301.

  1. the conclusion must be improved.

Reply: Thanks for the constructive comment. We have rewritten the implications of the study and the work to integrate this study with global interests. Correlation coefficients and annotated significance analysis for long and short-time scales were also added to the conclusions in Lines 394-427.

Round 2

Reviewer 1 Report

This manuscript can be accepted in its present form

Author Response

We would like to thank Reviewer #1 most sincerely for his/her constructive comments and valuable suggestions, which have helped us to thoroughly revise and improve the manuscript. Below please find our responses to Reviewer #1’s comments, and we have also incorporated them to the revised manuscript (RM).

1.This manuscript can be accepted in its present form

Reply: Thank you very much for your approval of our paper.

Reviewer 4 Report

The conclusion in the abstract are missing. 

Author Response

We would like to thank Reviewer #4 most sincerely for his/her constructive comments and valuable suggestions, which have helped us to thoroughly revise and improve the manuscript. Below please find our responses to Reviewer #4’s comments, and we have also incorporated them to the revised manuscript (RM).

  1. The conclusion in the abstract are missing.

Reply: Thanks for the constructive comment. We have revised the abstract in Lines 8-29.